# RBPGAN: RECURRENT BACK-PROJECTION GENERATIVE ADVERSARIAL NETWORK FOR VIDEO SUPER RESOLUTION

## ABSTRACT

In this paper, we propose a new Video Super Resolution algorithm in an attempt to generate videos that are temporally coherent, spatially detailed, and match human perception. To achieve this, we developed a new generative adversarial network named RBPGAN which is composed of two main components: a generator Network which exceeds other models for producing very high-quality frames, and a discriminator which outperforms others in terms of temporal consistency. The generator of the model uses a reduced recurrent back-projection network that takes a set of neighboring frames and a target frame, applies SISR (Single Image Super Resolution) on each frame, and applies MISR (Multiple Image Super Resolution) through an encoder-decoder Back-Projection based approach to concatenate them and produce x4 resolution version of the target frame. The Spatio-temporal discriminator uses triplets of frames and penalizes the generator to generate the desired results. Our contribution results in a model that outperforms earlier work in terms of perceptual similarity and natural flow of frames, while maintaining temporal coherence and high-quality spatial details.The algorithm was tested on different datasets to eliminate bias.

## 1 INTRODUCTION

Video Super Resolution (VSR) is the process of generating High Resolution (HR) Videos from Low Resolution (LR) Videos. Videos are of the most common types of media shared in our everyday life. From entertainment purposes like movies to security purposes like security camera videos, videos have become very important. As a result, VSR has also become important. There is a need to modernize old videos or enhance their quality for purposes such as identifying faces in security footages, enhancing satellite captured videos, and having a better experience watching old movies with today's quality.

Similar and older than VSR is ISR (Image Super Resolution), which is the process of generating a single high-resolution image from a single low-resolution image. Since a video is understood to be a sequence of frames (images), Video Super Resolution can be seen as Image Super Resolution (ISR) applied to each frame in the video. While this is useful because many of the ISR techniques can be slightly modified to apply to VSR, however, there are major differences between VSR and ISR. The main difference is the temporal dimension in videos that does not exist in images. The relationship between a frame in a video and other frames in the video is the reason why VSR is more complex than ISR (Haris et al., 2019).

In this research, various VSR methods will be explored. The methods are mainly clustered into two clusters, methods with alignment and methods without alignment (Liu et al., 2022). We will compare between the different methods across different datasets and discuss the results. Out of the methods we studied, we chose 2 models to be the base models for our research paper. We further explore these base models, experiment with them, and discuss possible enhancements. This paper aims to minimize the trade-off between temporal coherence, natural-to-the-eye perception, and quality of VSR. To achieve this, we propose a Generative Adversarial Network (GAN) that uses concepts from different state-of-art models and combines them in a way to achieve our goal.

Our methodology, experimentation and results are mentioned in this paper respectively. Finally, we conclude the paper and propose future work recommendations.

## 2 RELATED WORK

Based on our review of the literature, the Deep Learning based methods that target Video Super Resolution problem can be divided into 2 main categories: methods with alignment, and methods without alignment. Alignment basically means that the input LR video frames should be aligned first before feeding them into the model. Under the methods with alignment, existing models can be divided into two sub-categories: methods with Motion Estimation and Motion Compensation (MEMC), and methods with Deformable Convolution (DC). Under methods without alignment, existing models can be divided into 4 sub-categories: 2D convolution, 3D convolution, RNNs, and Non-Local based. In this section, the out-performing methods belonging to every category will be discussed.

### 2.1 METHODS WITH ALIGNMENT

#### 2.1.1 MOTION ESTIMATION AND MOTION COMPENSATION (MEMC)

First, the Temporally Coherent Generative Adversarial Network (TecoGAN) (Chu et al., 2020) The network proposes a temporal adversarial learning method for a recurrent training approach that can solve problems like Video Super Resolution, maintaining the temporal coherence and consistency of the video without losing any spatial details, and without resulting in any artifacts or features that arbitrarily appear and disappear over time. The TecoGAN model is tested on different datasets, including the widely used Vid4, and it is compared to the state-of-the-arts ENet, FRVSR, DUF, RBPN, and EDVR. TecoGAN is able to generate improved and realistic details in both down-sampled and captured images.However, one limitation of the model is that it can lead to temporally coherent yet sub-optimal details in certain cases such as under-resolved faces and text.
Second, the recurrent back-projection network (RBPN) (Haris et al., 2019). This architecture mainly consists of one feature extraction module, a projection module, and a reconstruction module. The recurrent encoder-decoder module integrates spatial and temporal context from continuous videos. This architecture represents the estimated inter-frame motion with respect to the target rather than explicitly aligning frames. This method is inspired by back-projection for MISR, which iteratively calculates residual images as reconstruction error between a target image and a set of its corresponding images. These residual blocks get projected back to the target image to improve its resolution. This solution integrated SISR and MISR in a unified VSR framework as SISR iteratively extracted various feature maps representing the details of a target frame while the MISR were used to get a set of feature maps from other frames. This approach reported extensive experiments for VSR and used different datasets with different specs to conduct detailed evaluation of its strength and weaknesses. This model reported significant results in terms of the quality of produced videos.

#### 2.1.2 DEFORMABLE CONVOLUTION METHODS (DC)

The Enhanced Deformable Video Restoration (EDVR) (Wang et al., 2019) is a framework that performs different video super-resolution and restoration tasks. The architecture of EDVR is composed of main modules known Pyramid, Cascading, and Deformable convolutions (PCD) and Temporal and Spatial Attention (TSA). The concept of deformable convolution is built on the idea of having sampled frames, augmenting their spatial locations, and learning additional offsets to these locations without supervision. This model results in very good results, however, it's size and number of parameters are very high.

### 2.2 METHODS WITHOUT ALIGNMENT

#### 2.2.1 2D AND 3D CONVOLUTION

2D convolution based VSR mainly uses the classic convolution method to extract information from the frames of the video in only the spatial dimension and increasing the resolution accordingly (Lu-

cas et al., 2019). However, 3d convolution is a better convolution method in which the temporal dimension is taken into consideration, produsing videos with better consistency. One high-performing method based on 3D convolution is Dynamic Upsampling Filters (DUF) (Jo et al., 2018). The dynamic filter network can generate filters that take specific inputs and generate corresponding features. The structure of the dynamic upsampling filter and the spatio-temporal information learned from the 3D convolution led to a comprehensive knowledge of the relations between the frames. However, a main drawback of this method is the time complexity.

### 2.2.2  RCNNS

RCNNS is a very powerful network (Dieng et al., 2019) developed a stochastic temporal convolutional network (STCN) by incorporating a hierarchy of stochastic latent variables into TCNs, allowing them to learn representations over a wide range of timescales. The network is divided into three modules: spatial, temporal, and reconstruction. The spatial module is in charge of extracting features from a series of LR frames. Temporal module is a bidirectional multi-scale convoluted version Motion estimation of LSTM that is used to extract temporal correlation between frames. The latent random variables in STCN are organized in accordance with the temporal hierarchy of the TCN blocks, effectively spreading them across several time frames. As a result, they generated a new auto-regressive model that combines the computational advantages of convolutional architectures with the expressiveness of hierarchical stochastic latent spaces. The model in STCN is meant to encode and convey information across its hierarchy.

### 2.2.3  NON-LOCAL METHODS

There is a progressive fusion network for VSR that is meant to make greater use of spatio-temporal information that has shown to be more efficient and effective than existing direct fusion, slow fusion, and 3D convolution techniques through a technique known as Progressive Fusion Video Super-Resolution Networks in Exploiting Non-Local Spatio-Temporal Correlations (PFNL). This is presented in Progressive Fusion Video Super-Resolution Network via Exploiting Non-Local Spatio-Temporal Correlations (Yi et al., 2019). That enhanced the non-local operation in this progressive fusion framework to circumvent the MEMC methods used in prior VSR techniques.This was done by adding a succession of progressive fusion residual blocks (PFRBs). The suggested PFRB is designed to make greater use of spatio-temporal information from many frames. So, the model can be summarized into three major components: a non-local resblock, progressive fusion residual blocks (PFRB), and an upsampling block. The non-local residual blocks are used to extract spatio-temporal characteristics, and PFRB is proposed to fuse them. Finally, the output of a sub-pixel convolutional layer is added to the input frame, which is then upsampled using bicubic interpolation to produce the final super-resolution results.

## 3  OUR MODEL AND CONTRIBUTION

After studying the previous work done in the area of VSR, we propose a Generative Adversarial Network that has a Recurrent Back Projection network as the generator to achieve highest possible accuracy without sacrificing the perceptual coherence.It offers superior results as it combines the benefits of the original MISR back-projection approach with Deep back-projection networks (DBPNs) which use the back-projection to perform SISR through estimating the SR frame using the LR frame.The discriminator of the model is inspired by the well-known TecoGAN paper to improve the temporal coherence of the generated frames. The architecture for the proposed architecture is shown in figure 1.

### 3.1  GENERATOR

The Recurrent Back Projection network basically calculates the residual images as reconstruction error between the target image and a set of neighboring images, 6 neighboring frames. It exploited temporal relationships between adjacent frames. The network mainly consists of three main modules, one feature extraction module, a projection module, and a reconstruction module. The feature extraction module basically performs two operations, it extracts the features directly from the target

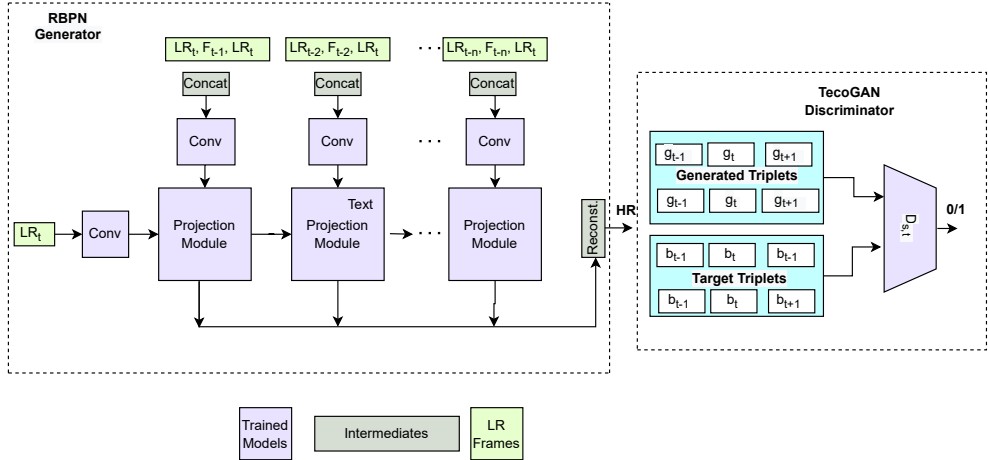

Figure 1: Proposed Architecture.

frame also extracts the feature from the concatenation of the neighboring frame, and calculates the optical flow from the neighboring frame to the target frame. The second module which is the projection module consists of an encoder and a decoder. The encoder is composed of a multiple image super-resolution (MISR), a single image super-resolution, and residual blocks. For the decoder, it consists of a strided convolution and a residual block. The decoder takes the output of the previous encoder as input to produce the LR features and then fed them to the encoder of the next projection module. The last module, which is the reconstruction module, takes the output of the encoder in each projection module and concatenates them as inputs to produce the final SR results.

## 3.2 DISCRIMINATOR

The discriminator, denoted by D, is Spatio-temporal based and its main role is to compare the generated HR frames from the generator with the ground truth.

This network also purposes a new loss function named "Ping-Pong" which focuses on the long-term temporal flow of the generated frames to make the results more natural without artifacts. The discriminator guides the generator to learn the correlation between the LR input and the HR targets. In addition, it penalizes the generator if the generated frames contain less spatial or unrealistic artifacts in comparison to the target HR and the original LR frames.

There is an issue that appears when super-resolving at large up-scaling factors and that is usually seen with CNNs Ledig et al. (2017). So, the proposed network chose this specific discriminator to mitigate the issue of a lack of finer texture details. The discriminator is trained to differentiate between the generated high resolution images and high resolution ground truth images.

The main goal of our proposed architecture is to recover precise photo-realistic textures and motion-based scenes from heavily down-sampled videos accordingly this improves the temporal coherence and perceptual quality.

## 4 DATASETS AND METRICS

### 4.1 DATASETS

We used the same datasets used by other VSR models for our results to be comparable to them. We used Vimeo-90k (Xue et al., 2019) and the training dataset created by TecoGAN publishers (we will refer to as VimeoTecoGAN) (Chu et al., 2020) for training experiments, and we used Vid4 and ToS3 for testing.You can see more details about the datasets in table 1.

We used 4 down-sampling scale with bicubic interpolation (also known as Gaussian Blur method) for training and testing, and we obtained comparable assessment findings on both testing datasets.

Table 1: Comparison between different Datasets

| Dataset | Resolutionn | Clips | Frames/clip | Frames |
|---------|-------------|-------|-------------|--------|
| Vimeo90K | $448x256$ | $13, 100$ | 7 | $91, 701$ |
| VimeoTecogan | Varies | 265 | 120 | $31, 800$ |
| Vid4 | Varies | 4 | Varies | 684 |
| ToS3 | $1280x534$ | 3 | $233, 166, 150$ | 549 |

## 4.2 EVALUATION METRICS

The usual criterion for evaluating the quality of super-resolution results mainly includes Peak signal-to-noise ratio (PSNR) and Structural index similarity (SSIM). PSNR is the ratio of an image's maximum achievable power to the power of corrupting noise that affects the quality of its representation. To calculate the PSNR of a picture, it must be compared to an ideal clean image with the highest potential power (ground truth). Higher outcomes are preferable. A single SR frame's PSNR can be calculated as

$$PSNR = 10log_10(\frac{MAX^2}{MSE})\tag{1}$$

where MAX is the color value's maximum range, which is commonly 255 and MSE is the mean squared error. SSIM measures the similarity of structure between two corresponding frames using an uncompressed or distortion-free image as a baseline. So, higher SSIM and PSNR values indicates higher quality. PSNR may be more sensitive to Gaussian noise, whereas SSIM may be more sensitive to compression errors. Their values, however, are incapable of reflecting video quality for human vision. That implies, even if a video has a very high PSNR value, it may still be unpleasant for humans. As a result, deep feature map-based measures like LPIPS (Zhang et al., 2018) can capture more semantic similarities. It indicates the extent to which the generated video is perceptually good and natural to the human-eye. The distance between picture patches is calculated using LPIPS (Learned perceptual image patch similarity). Higher implies more distinct. Lower values indicate a closer match. Additionally, ToF is used to calculate the pixel-wise difference of movements inferred from successive frames and it's a very indicative measure of the temporal coherence of video frames. The lower tOf, the more temporally consistent is the video.

$$tOF = ||OF(b_{t-1}, b_t) - OF(g_{t-1}, g_t)||\tag{2}$$

## 5 LOSS FUNCTIONS

The loss functions used while training our model are as follows:

1. GAN Loss (min-max loss)
   We use the Vanilla GAN loss (equation 3), which is the simplest form of the GAN loss, for the adversarial training. The generator tries to minimize it while the discriminator tries to maximize it.

   $$\min_G \max_D \mathbb{E}_{x \sim p_{\text{data}}(x)}[\log D(x)] + \mathbb{E}_{z \sim p_z(z)}[\log(1 - D(G(z)))]\tag{3}$$

   Here, $D(x)$ is the discriminator's estimate of the probability that real data instance x is real, and $D(G(z))$ is the discriminator's estimate of the probability that a fake instance is real. $\mathbb{E}$ is the expected value over all data instances.

2. Pixel loss
   Minimizes the pixel-wise squared differences between ground truth and generated frames (equation 4)

   $$||g_t - b_t||_2\tag{4}$$

3. Ping Pong Loss

Proposed by TecoGAN model (Chu et al., 2020), ping pong loss (equation 5) effectively avoids the temporal accumulation of artifacts, and targets generating natural videos that are consistent over time. PP loss uses a sequence of frames with the forward order as well as its reverse. Using an input number of frames of length n, we can form a symmetric sequence $a_1, ... a_{n-1}, a_n, a_{n-1}, ... a_1$ such that when feeding it to the generator, the forward results should be identical to the backward result .

$$\sum_{t=1}^{n-1} ||g_t - g_{t'}||_2 \tag{5}$$

Here, the forward results are represented with $g_t$ and the backward results with $g_{t'}$

4. Feature/perceptual Loss

represented by equation 6, it encourages the generator to produce features similar to the ground truth ones by increasing the cosine similarity of their feature maps. It ensures more perceptually realistic and natural generated videos. Our discriminator features contain both spatial and temporal information and hence are especially well suited for the perceptual loss.

$$1 - \frac{\Phi(I_{s,t}^g) * \Phi(I_{s,t}^b)}{||\Phi(I_{s,t}^g)|| * ||\Phi(I_{s,t}^b)||} \tag{6}$$

Where $I^g = \{g_{t1}, g_t, g_{t+1}\}, I^b = \{b_{t1}, b_t, b_{t+1}\}$

5. Warping Loss

Used while training the motion estimation network ($F$) that produces the optical flow between consecutive frames.

$$\sum ||a_t - W(a_{t-1}, F(a_{t-1}, a_t))||_2 \tag{7}$$

equation 7 represents warping loss, where $W()$ is the warping function, $F()$ is the flow estimator, and $a_t$ is the LR frame in position $t$.

## 6 EXPERIMENTS

In all our experiments, we focus on the 4× Super Resolution factor since it provides satisfactory results and takes a reasonable amount of training. Also, we used crop size of 32x32 and gaussian downsampling. All experiments were done using following specifications to enable the dense nature of the training phase: 64GB of DDR4 RAM, 2.80GHz Intel Core i9-10900F CPU, NIVIDIA GetForce RTX 3090 (1 x 24 GB) GPU, and Ubuntu 20.04.3 LTS operating system.

We will now present and explain the experiments we did in sequence, and later we will explain and discuss their results comparatively.

So, we started by training and testing our two base models (TecoGAN and RBPN) to ensure their correctness and reliability before integrating them and produce our model. Then, we integrated them as discussed in section 2. Later, we performed some experiments on our model with different parameters and loss functions till we reached the best outcome. The final model is later compared with the other state-of-the-art models in terms of PSNR, SSIM, LPIPS, and ToF metrics

### 6.1 EXPERIMENT 1: REDUCED RBPN MODEL SIZE

As discussed, RBPN is the base model we are using for our model's generator, and we started by training and testing it. The model size was very large, leading to high reference and training time, and therefore we decided to build our generator based on a reduced size of a recurrent back-projection network. We experimented with the number of neighbour frames passed to the projection modules in the model and decreased it without compromising much of the produced quality. This

Table 2: Comparative analysis between all the conducted experiments on our model for **Vid4** dataset

| Metric Name | Experiment 3.1 | Experiment 3.2 | Experiment 3.3 |
|:---:|:---:|:---:|:---:|
| PSNR | 25.58 | **25.74** | 25.56 |
| LPIPS | 1.47 | **1.44** | 1.45 |
| tOF | 2.46 | **2.35** | 2.40 |
| SSIM | 0.756 | **0.762** | 0.751 |

resulted in a decreased number of layers and smaller model. The training of this experiment took around 1 hour/epoch and we trained it for 150 epochs, using VimeoTecoGAN dataset and the loss functions mentioned in section 5.

## 6.2 EXPERIMENT 2: MODEL INTEGRATION

After we ensured the correctness, reliability and readiness of the two base models for the integration phase, we began integrating the generator with the spatio-temporal discriminator to create our GAN model and prepare it for some experiments. The integration was a challenging task since the two models had many differences in functions' interfaces, dependencies used, training datasets and the lack of generalization to fit any other dataset.

We will now discuss the experiments done on RBPGAN (our model) to monitor the model potential and test our hypothesis.

### 6.2.1 EXPERIMENT 3.1

We used all loss functions mentioned in the previous section (Ping Pong loss, Pixel loss, Feature loss, Warping loss, and GAN loss). We used 2 neighbour (adjacent) frames per target frame, but due to the use of ping pong loss, this number is doubled to create the backward and forward paths. Therefore, the generator was using 4 neighbour frames per frame. We trained both the generator and discriminator simultaneously from the beginning, using the VimeoTecoGAN dataset, and training took around 3.5 days using the specs mentioned.

### 6.2.2 EXPERIMENT 3.2

We used the same loss functions as experiment 3.1 except the ping pong loss to observe its effect on the results. We used 3 neighbour frames per target frame, started the training of generator and discriminator together, and used the same dataset and other parameters as experiment 3.1. The training took around 3 days.

### 6.2.3 EXPERIMENT 3.3

This experiment is the same as experiment 3.2 except that we firstly trained the generator solely for a number of epochs and then started the training of the GAN using this pre-trained part. The training took aound 3 days using the same dataset, number of neighbours and other parameters.

### 6.2.4 EXPERIMENT 4

We trained RBPN model with the same number of neighbours, crop size, dataset, and other unifyable parameters as we did for our model in the 3 previous experiments to ensure fair comparison between it and our model.

## 7 RESULTS

Following are the results and metrics evaluation for the experiments done and explained in previous section (tables 2 and 3):

Table 3: Comparative analysis between all the conducted experiments on our model for **ToS3** dataset

| Metric Name | Experiment 3.1 | Experiment 3.2 | Experiment 3.3 |
|---|---|---|---|
| PSNR | **32.89** | 32.85 | 32.78 |
| LPIPS | 0.78 | **0.69** | 0.75 |
| tOF | **1.60** | 1.64 | 1.62 |
| SSIM | 0.872 | **0.880** | 0.869 |

So, overall, experiment 3.2 yields the best results collectively, and therefore we will use it to compare with the state-of-the-art models (tables 4 and 5). Also, you see in figures 2 and 3 some examples from vid4 dataset for our model.

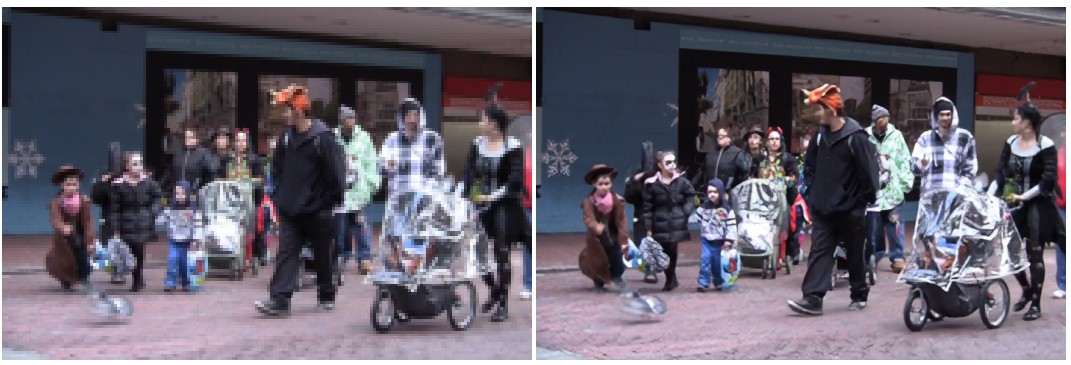

Figure 2: walk (Left:LR , Right: RBPGAN)

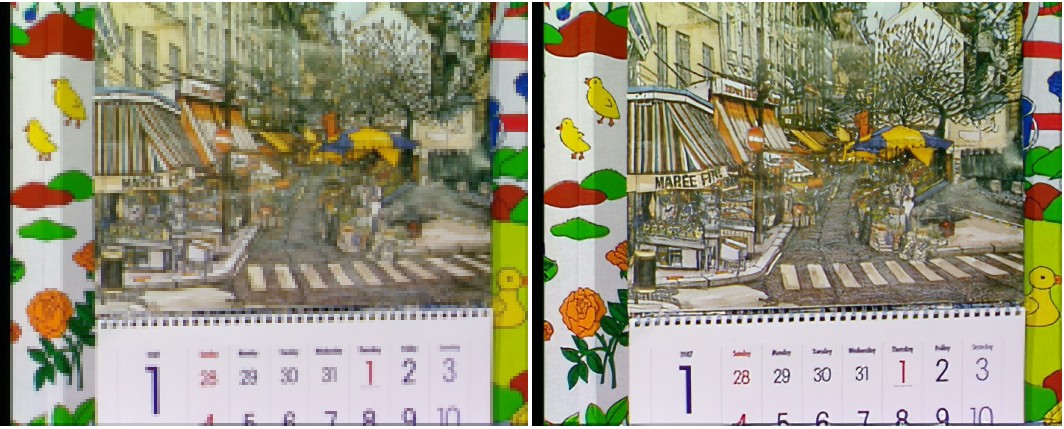

Figure 3: calendar (Left:LR , Right: RBPGAN)

Table 4: Comparison between experiment 2 and state-of-the-arts for **Vid4** dataset

| Metric | Experiment 3.2(Ours) | TecoGAN | RBPN (3 neighbors) | BIC | ENet | DuF |
|---|---|---|---|---|---|---|
| PSNR | 25.74 | 25.57 | 26.71 | 23.66 | 22.31 | **27.38** |
| LPIPS | **1.44** | 1.62 | 2.0 | 5.04 | 2.46 | 2.61 |
| tOF | 2.35 | 1.90 | 2.19 | 5.58 | 4.01 | **1.59** |
| SSIM | **0.756** | 0.770 | 0.801 | NA | NA | 0.815 |

Table 5: Comparison between experiment 2 and state-of-the-arts for **ToS3** dataset

| Metric | Experiment 3.2(Ours) | TecoGAN | RBPN (3 neighbors) | BIC | ENet | DuF |
|--------|----------------------|---------|--------------------|-----|------|-----|
| PSNR | 32.85 | 32.65 | 34.32 | 29.58 | 27.82 | **34.6** |
| LPIPS | **0.69** | 1.09 | 1.10 | 4.17 | 2.40 | 1.41 |
| tOF | 1.64 | 1.34 | 1.54 | 4.11 | 2.85 | **1.11** |
| SSIM | **0.880** | 0.892 | 0.915 | NA | NA | NA |

## 8    DISCUSSION AND CONCLUSION

We can say that we managed to prove our hypothesis and achieved the highest results when it comes to perceptual similarity, and very high results in temporal coherence and quality of frames. While current adversarial training produces generative models in a variety of fields, temporal correlations in the generated data have received far less attention. We, on the other hand, concentrated on improving learning goals and present a temporally self-supervised approach to bridge this gap. For the sequential generation tasks such as video super-resolution and unpaired video translation, natural temporal shifts are critical. The reduced Recurrent Back-Projection Network in our model treats each context frame as a separate source of information. These sources are combined in an iterative refinement framework inspired by the idea of back-projection in multiple-image super-resolution. This is aided by explicitly representing estimated inter-frame motion with respect to the target, rather than explicitly aligning frames. In this work, we were trying to merge and advance the highly realistic output of RBPN with the more naturally-looking output of TecoGAN to have a smooth high quality generated videos from our network. In the results section, we demonstrated the metrics of generated frames and how we achieve higher qualities, represented by SSIM and PSNR, when compared with TecoGAN. We also maintained high temporal consistency, represented by tOf, compared with all the models. When assessing our model with the LPIPS metric, which is responsible for measuring the perceptual similarity, we found that our model surpasses all models, including our two base models, TecoGAN and RBPN. This is because the discriminator helps the generator in learning further the correlations between sequential frames. Adding to that, we managed to create the model with a reasonable size and number of trained parameters compared to the other powerful models such as DuF, while producing distinguishing results.

There are some limitations we faced during the implementation; our model needs powerful computational resources but we were only limited to a 2-GPU machine with a 64GB memory which limited our experiments since each experiment took longer than usual on the available machines. Moreover, while our technique produces extremely realistic results for a large variety of natural pictures, in some circumstances, such as under-resolved faces and text in VSR, or jobs with dramatically differing motion between two domains, our method can provide temporally coherent and natural but sub-optimal details. It would be interesting to use both our technique and motion translation from contemporaneous work in the latter instance  (Wang et al., 2019). Hence, we recommend using different downsampling methods to introduce more generalization to the model, and may be train the model on more augmented datasets. In addition, there is continuing research on the usage of accelerator techniques to boost the speed of network training and inference time, leading to real-time VSR transition.

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
