# OpenReview forum: "Recurrent Back-Projection Generative Adversarial Network for Video Super Resolution"
_ICLR.cc/2023/Conference — Submitted to ICLR 2023_

### Official Review · Reviewer_LUmB · 2022-10-23

**Confidence:** 3
**Correctness:** 2
**Technical Novelty And Significance:** 1
**Empirical Novelty And Significance:** 1
**Recommendation:** 1

**Clarity, Quality, Novelty And Reproducibility:**

Clarity: Poor

Quality: Poor

Novelty: Poor

Reproducibility: Good, as it is just the combination of two methods.

**Strength And Weaknesses:**

Weaknesses:
1) The paper is poorly written. Many typos can be found in the paper. For example, in Fig. 1, there is a "text" appearing in the box of "Projection Module".
2) The proposed method is just a combination of existing methods, i.e., RBPN and TecoGAN. It is too trivial.
3) The performance is not compared with existing VSR methods except RBPN and TecoGAN. The comparisons with RBPN and TecoGAN should be regarded as a part of the ablation study.

**Summary Of The Paper:**

The paper combines RBPN and TecoGAN to solve the Video Super-Resolution task. Experiments show that the performance is better than RBPN and TecoGAN.

**Summary Of The Review:**

The paper is of poor quality. The idea is a combination. The experiments are not thorough and convincing enough.

---

### Official Review · Reviewer_jYRg · 2022-10-24

**Confidence:** 5
**Clarity, Quality, Novelty And Reproducibility:** 1. The writing of this paper should b…
**Correctness:** 2
**Technical Novelty And Significance:** 1
**Empirical Novelty And Significance:** 1
**Recommendation:** 1

**Strength And Weaknesses:**

Strength: This paper proposes a new GAN-based VSR method to address the important issues of temporally coherent, spatially details and human perception.

Weaknesses: The writing of this paper should be improved. The details of the proposed method are not clear. The experiment section should be improved.

**Summary Of The Paper:**

This paper proposes a new Video Super Resolution (VSR) method based on generative adversarial networks (called RBPGAN) to improve temporally coherent, spatially details, and human perception. In particular, the generator uses a reduced recurrent back-projection network, and the discriminator measures the temporal consistency. Some experiments are provided to demonstrate the effectiveness of the proposed method.

**Summary Of The Review:**

The writing should be improved, the technical details and more experiments should be provided.

---

### Official Review · Reviewer_6Vz2 · 2022-10-24

**Confidence:** 4
**Correctness:** 2
**Technical Novelty And Significance:** 2
**Empirical Novelty And Significance:** Not applicable
**Recommendation:** 3

**Clarity, Quality, Novelty And Reproducibility:**


This paper clearly describes the strong motivation, current challenge of the problem, and the necessity of proposing such a new method.
This paper has provided a comprehensive literature review, covering all the campaigns in the video super resolution community.

However, the proposed method section is too short and the description is not quite informative.
The clarity of the paper is not quite good.  Since the description for the method itself is quite confusing, it is difficult to judge the originality of the work.


**Strength And Weaknesses:**

Strength
1. This paper provides a comprehensive literature review.


Weakness
1. In section 3.1 generator , the described encoder and decoder cannot be located in Figure 1. It is quite confusing how the proposed method arrange the SISR and MISR modules in the network.
2. The Figure 1  does not provide the details of each projection module. By reading the figure and the method section, it is difficult to understand how the network should work.
3. In Section 3, it does not validate the reason why the proposed network should work and to what extent it works.


4. There are many typos in the submitted manuscript.
(1). A comma is missing in Eq.(1) . The log with base 10 is not properly typed.
(2). Almost all the equations in the article are missing punctuations.
(3). Sentence after Eq.(5): Here the forward results are represented with g_t -> represented by g_t.
(4). 4th point of Section 5 Loss functions, The first sentence is not properly written and formatted.
(5). Section 6 Experiments, first paragraph, the past tense and present tense are mixed together.
(6). The caption of Figure 1 should come with : The proposed architecture.

**Summary Of The Paper:**

This paper proposes a new CNN-based method that deals with video super-resolution problems. The proposed method consists of a generator and a discriminator. The generator combines Single Image Super-Resolution and Multiple Image super-Resolution on target frame and its neighboring frames. The discriminator uses triplet loss to enhance the capability of discriminating scaled output from the generator. Experiments show that the proposed method outperforms other baselines.



**Summary Of The Review:**

In summary, this paper fails to clearly describe the proposed method combining both texts and figures. Hence, it is quite difficult to judge the validity and the effectiveness of the work. The authors are suggested to provide a detailed explanation of the proposed method and also to give reasons of the network design.

---

### Official Review · Reviewer_aKDa · 2022-11-04

**Confidence:** 5
**Correctness:** 1
**Technical Novelty And Significance:** 1
**Empirical Novelty And Significance:** 1
**Recommendation:** 1

**Clarity, Quality, Novelty And Reproducibility:**

Writing quality is not very good and there are many typos, latex grammar errors.
Novelty is poor, there are no new ideas or techniques.
Implementation details are not described in detail.


**Details Of Ethics Concerns:**

The authors bring sentences from one of the previous work, RBPN.
I doubt this is a mistake or coincidence.

Please refer to the strengths & weaknesses.

**Strength And Weaknesses:**

- Novelty

There is not much novelty in this work.
The generator architecture is from RBPN, discriminator is from TecoGAN, the loss functions are widely used popular terms.

- Experimental Results

While the authors claim they achieve state-of-the-art performance in perceptual quality, there is no visual comparison with the other methods. In Table 5, the proposed method achieves good LPIPS and SSIM, however, I believe this is not a fair comparison. There is trade-off relation between fidelity-based metrics and perceptual quality metrics. Such perception-distortion trade-off is well known in image/video restoration community. The proposed method far lacks in PSNR.

Y. Blau and T. Michaeli, “The Perception-Distortion Tradeoff,” CVPR 2018



- Concerns

The authors directly copy the sentences from RBPN paper.
For example, in section 8,

“The reduced Recurrent Back-Projection Network in our model treats each context frame as a separate source of information. These sources are combined in an iterative refinement framework inspired by the idea of back-projection in multiple-image super-resolution. This is aided by explicitly representing estimated inter-frame motion with respect to the target, rather than explicitly aligning frames”

This is an except from the abstract of Haris et al., 2019. This is considered plagiarism.



**Summary Of The Paper:**

This paper trains RBPN model for video super-resolution. The authors employ TecoGAN discriminator and their ping-pong loss to improve temporal consistency of the super-resolved video. The authors use perceptual loss and warping loss, too.


**Summary Of The Review:**

I don’t find many reasons to accept this paper.
The paper lacks novelty and the experimental results are hard to claim the benefits over the previous methods.
Furthermore, I seriously concern about plagiarism the issue. The authors copied sentences from the previous work which they bring the core architecture.

---

### Decision · Program_Chairs · 2023-01-20

**Decision:**

Reject

**Justification For Why Not Higher Score:**

No basis

**Justification For Why Not Lower Score:**

N/A

**Metareview: Summary, Strengths And Weaknesses:**

Uniformly negative reviews, pointing out very limited technical (or other) novelty, issues in presentation and clarity, and in one of the reviews, potential plagiarism (or self-plagiarism). I have looked at the latter and in my opinion no special treatment is needed; while some text does appear to be taken verbatim from RBPN, it is not extensive and likely the result of authors' ignorance about the unacceptable nature of such practice, rather than an attempt to deceive. The authors should be clearly informed about this and admonished to avoid such practice in the future.

On the merits, the paper is clearly below threshold, and sans rebuttal, I agree that it should not be accepted.